# Factors that contribute to loss to follow-up in the medium term after initiation of anti-vascular endothelial growth factor therapy for neovascular age-related macular degeneration in Japanese patients

Takaaki Sugisawa[1¶], Fumi Gomi[1¶*], Yuri Harada[2¶], Hiroko Imaizumi[3¶], Shuichiro Aoki[3¶], Akiko Miki[4¶], Maya Kishi[4¶], Tomofusa Yamauchi[5¶], Daisuke Nagasato[5¶], Yoko Ozawa[6,7¶], Masatoshi Haruta[8¶], Nobuhiro Kato[8¶], Hisashi Matsubara[9¶], Tsutomu Yasukawa[10¶], Aki Kato[10¶], Hiroto Terasaki[11¶], Takao Hirano[12¶], Yasuhiro Iesato[12¶], Hiroki Tsujinaka[13¶], Tomoya Murakami[14¶], Yoshinori Mitamura[15¶], Makiko Wakuta[16¶], Kazuhiro Kimura[16¶], Masahiko Shimura[17¶], on behalf of the J-CREST (Japan Clinical Retina Study) Group[¶]

1 Department of Ophthalmology, Hyogo Medical University, Nishinomiya, Japan, 2 Shiga University of Medical Science, Ōtsu, Japan, 3 Department of Ophthalmology, Sapporo City General Hospital, Sapporo, Japan, 4 Department of Surgery, Division of Ophthalmology, Kobe University Graduate School of Medicine, Kobe, Japan, 5 Department of Ophthalmology, Saneikai Tsukazaki Hospital, Himeji, Japan, 6 Department of Clinical Regenerative Medicine, Fujita Medical Innovation Center Tokyo, Eye Center, Fujita Health University, Haneda Clinic, Tokyo, Japan, 7 Department of Ophthalmology, Keio University School of Medicine, Tokyo, Japan, 8 Department of Ophthalmology, Kurume University School of Medicine, Kurume, Japan, 9 Department of Ophthalmology, Mie University Graduate School of Medicine, Tsu, Japan, 10 Department of Ophthalmology and Visual Science, Nagoya City University Graduate School of Medical Sciences, Nagoya, Japan, 11 Department of Ophthalmology, Kagoshima University Graduate School of Medical and Dental Science, Kagoshima, Japan, 12 Department of Ophthalmology, Shinshu University School of Medicine, Matsumoto, Japan, 13 Department of Ophthalmology, Nara Medical University, Kashihara, Japan, 14 Department of Ophthalmology, Faculty of Medicine, University of Tsukuba, Tsukuba, Japan, 15 Department of Ophthalmology, Tokushima University Graduate School, Tokushima, Japan, 16 Department of Ophthalmology, Yamaguchi University School of Medicine, Yamaguchi, Japan, 17 Department of Ophthalmology, Tokyo Medical University Hachioji Medical Center, Tokyo, Japan

¶Membership of the J-CREST (Japan Clinical Retina Study) Group (The complete list of authors of this group can be found in the Acknowledgement).
* gomi.fumi@gmail.com

## Abstract

### Purpose

To identify time-specific factors associated with loss to follow-up (LTFU) in the early to medium term after initiating anti-vascular endothelial growth factor (VEGF) treatment in patients with neovascular age-related macular degeneration (nAMD) in Japan.

### Methods

The study had a retrospective multicenter case–control design and was performed across 16 specialist retinal facilities in Japan. Patients diagnosed with nAMD at their initial visit who initiated treatment between January 2017 and December 2020 were

**Data availability statement:** All relevant data are stored exclusively on my personal computer, located in a locked cabinet within the Hyogo Medical University . Due to security and confidentiality requirements, the data are not stored online or on any public repository. Researchers who meet the criteria for access to confidential data may request access via the Hyogo Medical University Institutional Data Access Committee (contact via [ta-sugisawa@hyo-med.ac.jp]).

**Funding:** The author(s) received no specific funding for this work.

**Competing interests:** NO authors have competing interests.

included. Patient characteristics were analyzed to identify factors associated with LTFU within 3 months (very early), 3 months to 1 year (early), and 1 year to 2 years (medium term) after starting treatment.

## Results

Data for 2389 patients with nAMD were analyzed. The very early, early, and medium-term LTFU rates were 6.8%, 13.8%, and 21.2%, respectively. Stepwise regression analysis identified factors that were significantly associated with LTFU at a very early stage to be greater central retinal thickness at baseline and a prior treatment history, those associated with early LTFU to be worse baseline best-corrected visual acuity (BCVA), anti-VEGF treatment combined with photodynamic therapy, and a follow-up period that overlapped with the COVID-19 pandemic, and that associated with medium-term LTFU to be worse BCVA at 3 months. LTFU in any period within 3 months to 2 years was more likely in patients aged >80 years, and LTFU very early within 3 months was more likely in those aged <60 years. A poor baseline BCVA (logMAR) of >1 was a risk factor for LTFU within 3 months and 1 year, whereas LTFU was significantly less likely in patients with good baseline BCVA (<0.1).

## Conclusion

The LTFU rate in patients with nAMD increased over time. Factors contributing to LTFU vary depending on the time since initiation of treatment.

## Introduction

Neovascular age-related macular degeneration (nAMD) is a debilitating disease that affects visual function in the elderly. The prognosis of nAMD has improved dramatically since the advent of anti-vascular endothelial growth factor (VEGF) injections [1,2]. Over the past decade, various pharmacological agents have become available in Japan and are widely used in healthcare facilities. A number of clinical studies have demonstrated the ability of regular anti-VEGF injections to improve vision over the long term in patients with nAMD in Japan [3,4].

Interruption or discontinuation of treatment for nAMD can lead to deteriorating vision [5] and may result in visual impairment that is irreversible even after resumption of treatment [6]. Therefore, it is important to understand the factors contributing to discontinuation of anti-VEGF treatment in each region and implement appropriate measures to mitigate rates of loss to follow-up (LTFU). The aim of this study was to determine the rate of LTFU in patients receiving anti-VEGF therapy for nAMD across 16 specialist retinal facilities in Japan during the first 3 years of treatment and to identify the time-specific risk factors associated with LTFU.

## Materials and methods

The study had a multicenter retrospective observational design and was approved by the institutional review boards of all participating hospitals and conducted in

accordance with the Declaration of Helsinki. The study protocol was approved by the ethics committee at the principal investigator's institution (Hyogo Medical University, protocol number 3758). The article follows the reporting guidelines for a non-randomized case series. The requirement for informed consent was waived in view of the retrospective observational nature of the study. The hospital database was accessed, and the data between 31 July and 31 December 2021 were collected. The patient data were fully anonymized for analysis. The authors did not have access to information that could identify individual participants during or after data collection.

The study included consecutive patients with nAMD in whom treatment with an anti-VEGF agent was initiated between January 1, 2017 and December 31, 2020 at any of 16 member institutions of the Japan Clinical Retina Study Group (J-CREST). These institutions are specialized in retinal care and receive patients referred from local clinics for treatment. The final data were collected on March 31, 2021. The following data were extracted from the medical records for patients who had follow-up durations within 3 months, 1 year, and 2 years during the study period: age, sex, whether nAMD was unilateral or bilateral, medical history, best-corrected visual acuity (BCVA), central subfoveal thickness (CST, distance between the inner limiting membrane and retinal pigment epithelium), subtype of disease (typical nAMD, polypoidal choroidal vasculopathy (PCV), or retinal angiomatous proliferation (RAP)), initial treatment modality at each facility (anti-VEGF therapy alone or combined with photodynamic therapy [PDT], or other), and number of anti-VEGF injections and visits. All data were anonymized for analysis. When both eyes were diagnosed with nAMD at baseline, the eye that developed nAMD later was included as the assigned eye for the primary analysis. BCVA was recorded using the Landolt C chart and converted to the logarithm of the minimum angle of resolution (logMAR) for statistical analysis.

Figure 1 shows the period for patients' inclusion for the respective follow-up duration period by the date of treatment initiation. The start of the COVID-19 pandemic was defined as March 1, 2020, when COVID-19 started to spread widely in Japan. When the follow-up period overlapped by at least one month with the COVID pandemic. the patients involved were defined as "after COVID-19".

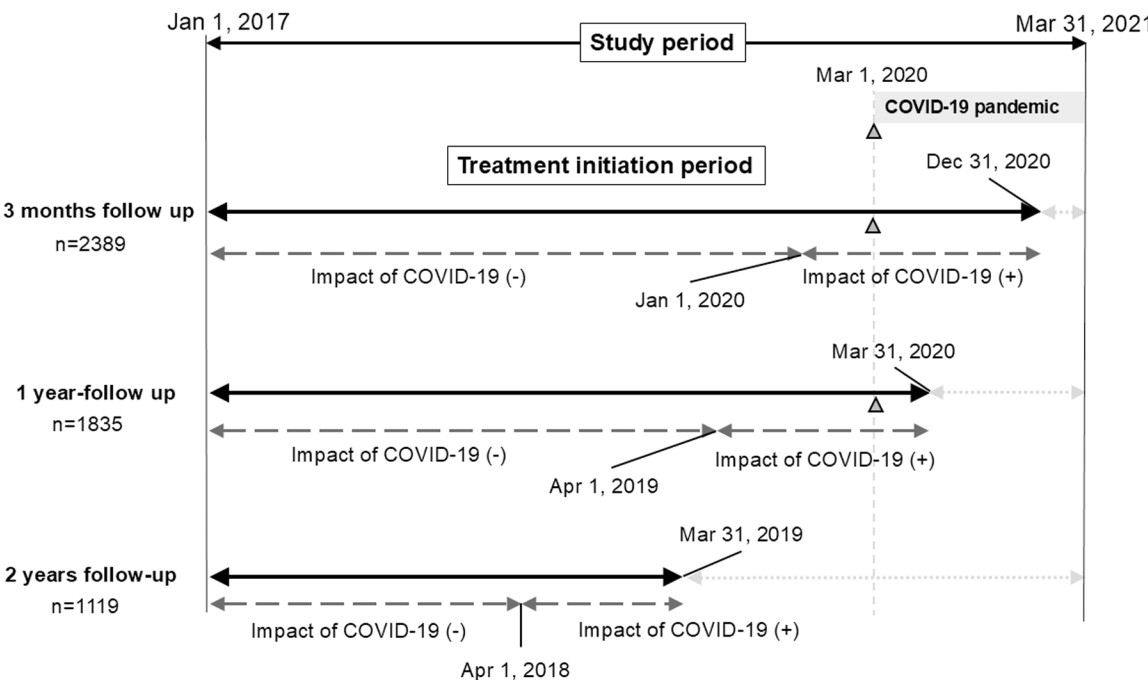

**Fig 1. Patients' inclusion period for analyzing follow-up status for up to 3 months, 3 months to 1 year, and 1 year to 2 years.**

In this study, patients who did not attend for scheduled examinations beyond one month from the appointed date and those who transitioned to follow-up at a local clinic were defined as LTFU. Factors contributing to LTFU within 3 months, 3 months to 1 year, and 1 year to 2 years were examined in univariate analyses. A categorical evaluation was performed for patient age, baseline BCVA, and disease subtype to determine whether they affected the likelihood of LTFU.

## Statistical analysis

Continuous data were compared between non-paired groups using the Student's *t*-test or the Wilcoxon test, and categorical data were compared using the chi-squared test. Continuous variables are presented as the mean±standard deviation. The statistical significance of differences in qualitative variables was evaluated using Fisher's exact test. Patients with missing data were excluded. For variables with a p-value of <0.05 in univariate analysis, a stepwise regression analysis method was applied using the minimum Bayesian information criterion as the selection criterion to identify the most relevant variables associated with the outcome variable. Finally, multivariable logistic regression analysis was used to assess the variables associated with LTFU. The analysis shows the odds ratios (ORs) associated with LTFU together with their 95% confidence intervals (CIs). All statistical analyses were performed using JMP Pro 15 (SAS Institute Inc., Cary, NC, USA). A p-value <0.05 was considered statistically significant.

## Results

The total number of patients analyzed was 2421, and the numbers of patients available for the follow-up analysis within 3 months, 3 months to 1 year, and 1 year to 2 years were 2389, 1835, and 1119, respectively. The initial anti-VEGF agents used in this study were ranibizumab (17%) and aflibercept (83%). No remarkable differences in the patient characteristics at baseline or during the follow-up period were observed among the 16 institutions. However, there was a significant difference in the choice of initial anti-VEGF agent among the institutions (p<0.0001).

### Comparison of the characteristics of patients who were LTFU and those who continued follow-up by time after treatment initiation

Table 1 compares the characteristics of patients who were LTFU within 3 months after initiation of treatment with those in patients who continued follow-up during this time. The LTFU rate within the first 3 months of treatment was 6.8%. Patients

**Table 1. Characteristics of patients with neovascular age-related macular degeneration who were lost to follow-up and those of patients who continued follow-up for 3 months after initiating anti-VEGF treatment.**

|  | Lost to follow-up | Continued follow-up | p-value |
|---|---|---|---|
| Number | 162 | 2227 |  |
| Age, years, mean±SD | 79.9±10.4 | 75.3±9.2 | *0.013* |
| Female sex (%) | 34 | 32 | 0.54 |
| Prior treatment history (%) | 20 | 11 | *<0.001* |
| Type of nAMD (typical/PCV/RAP) | 43.1/48.8/8.1 | 45.9/48.6/6.4 | 0.67 |
| Bilateral involvement (%) | 13.7 | 21.2 | *0.022* |
| BCVA, logMAR, mean±SD, assigned eye | 0.54±0.46 | 0.40±0.42 | *<0.001* |
| BCVA, logMAR, mean±SD, fellow eye | 0.17±0.46 | 0.15±0.43 | 0.6 |
| CST, µm, mean±SD | 442±228 | 373±167 | *<0.001* |
| Combined with PDT (%) | 10.8 | 8.4 | 0.28 |
| After COVID-19 pandemic | 24.1 | 23.5 | 0.87 |

nAMD, neovascular age-related macular degeneration; BCVA, best-corrected visual acuity; CST, central subfoveal thickness; logMAR, logarithm of the minimum angle of resolution; PCV, polypoidal choroidal vasculopathy; PDT, photodynamic therapy; RAP, retinal angiomatous proliferation; SD, standard deviation; VEGF, vascular endothelial growth factor.

who were LTFU by 3 months were older (p = 0.013) and more likely to have a prior treatment history (p < 0.001), a low likelihood of bilateral involvement (p = 0.022), worse baseline BCVA (p < 0.001), and greater CST (p < 0.001) at baseline.

Table 2 compares the characteristics of patients who were LTFU between 3 months and 1 year after initiation of treatment and those who continued follow-up during this time. The LTFU rate between 3 months and 1 year after initiation of treatment was 13.8%. Patients who were lost to follow-up by 1 year were older at the start of treatment (p < 0.001), had a prior treatment history (p = 0.038), had worse BCVA in the assigned eye (p < 0.001), were more likely to be receiving adjunctive PDT (p = 0.028), had worse visual acuity at 3 months after initiation of treatment (p < 0.001), and started treatment after the COVID-19 pandemic (p = 0.0036).

Table 3 compares the characteristics of the patient group that was LTFU between 1 year and 2 years after initiation of treatment and who continued follow-up during this time. The LTFU rate between 1 year and 2 years was 21.2%. Patients who were lost to follow-up within 2 years were older at initiation of treatment (p = 0.007), were more likely to be male (p = 0.024), had worse BCVA in the assigned and fellow eye (p < 0.001 and p = 0.0079, respectively), worse BCVA and CST at 3 months (p < 0.001 and p = 0.049), and worse BCVA and CST at 1 year (p < 0.001 and p = 0.0045), and were likely to have received fewer injections during the first year (p = 0.013).

## Comparison of the reasons for LTFU between 3 months, 1 year, and 2 years after treatment initiation

Table 4 shows the results of the stepwise regression analysis using risk factors for LTFU identified in univariate analyses for the respective durations after treatment initiation. Significant reasons for LTFU within 3 months were greater CST at the initial visit and a prior treatment history. Logistic regression analysis of each factor revealed an OR of 1.002 (95% CI 1.001–1.003) for greater CST at baseline and 2.41 (95% CI 1.56–3.71) for prior treatment history. Significant reasons for LTFU between 3 months and 1 year after treatment initiation were worse BCVA at baseline, combination treatment with PDT, and a follow-up period that overlapped with the COVID-19 pandemic. Logistic regression analysis revealed ORs

**Table 2. Characteristics of patients with neovascular age-related macular degeneration who were lost to follow-up and those of patients who continued follow-up for one year after initiating anti-VEGF treatment.**

|  | Lost to follow-up | Continued follow-up | p-value |
|---|---|---|---|
| Number | 253 | 1582 |  |
| Age, years, mean ± SD | 76.8 ± 10.3 | 75.0 ± 9.0 | *<0.001* |
| Female sex (%) | 32 | 31 | 0.77 |
| Prior treatment history | 14 | 9.7 | *0.038* |
| Type of nAMD (typical/PCV/RAP) | 45.8/48.6/5.6 | 44.4/49.4/6.3 | 0.86 |
| Bilateral involvement (%) | 17.9 | 22.3 | 0.12 |
| BCVA, logMAR, mean ± SD, assigned eye | 0.51 ± 0.49 | 0.37 ± 0.40 | *<0.001* |
| BCVA, logMAR, mean ± SD, fellow eye | 0.17 ± 0.42 | 0.15 ± 0.44 | 0.069 |
| CST, μm, mean ± SD | 396 ± 190 | 367 ± 158 | 0.19 |
| Combined with PDT (%) | 12.5 | 8.2 | *0.028* |
| After COVID-19 pandemic | 38% | 29% | *0.0036* |
| BCVA at 3M, logMAR, mean ± SD, assigned eye | 0.40 ± 0.47 | 0.28 ± 0.38 | *<0.001* |
| CSTat 3M, μm, mean ± SD | 266 ± 123 | 251 ± 102 | 0.35 |
| Changes in visual acuity between initial visit and 3M, mean ± SD | −0.11 ± 0.27 | −0.089 ± 0.26 | 0.53 |
| Changes in CST between initial visit and 3M, μm, mean ± SD | −122 ± 161 | −114 ± 139 | 0.91 |

3M, 3 months; nAMD, neovascular age-related macular degeneration; BCVA, best-corrected visual acuity; CST, central subfoveal thickness; ILM–RPE, inner limiting membrane–retinal pigment epithelium; logMAR, logarithm of the minimum angle of resolution; PCV, polypoidal choroidal vasculopathy; PDT, photodynamic therapy; RAP, retinal angiomatous proliferation; SD, standard deviation; VEGF, vascular endothelial growth factor.

**Table 3. Characteristics of patients with neovascular age-related macular degeneration who were lost to follow-up and those of patients who continued follow-up for 2 years after initiating anti-VEGF treatment.**

| Initiation date to March 31, 2019 | Lost to follow-up | Continued follow-up | p-value |
|---|---|---|---|
| Number | 237 | 882 | |
| Age, years, mean±SD | 76.6±9.7 | 74.7±8.7 | <0.001 |
| Female sex (%) | 24.9 | 32.5 | 0.024 |
| Prior treatment history | 8.9 | 8.9 | 0.99 |
| Type of nAMD (typical/PCV/RAP) | 43.0/51.5/5.5 | 44.6/48.8/6.7 | 0.67 |
| Bilateral involvement (%) | 21.1 | 23.5 | 0.44 |
| BCVA, logMAR, mean±SD, assigned eye | 0.45±0.41 | 0.36±0.40 | <0.001 |
| BCVA, logMAR, mean±SD, fellow eye | 0.20±0.46 | 0.14±0.44 | 0.0079 |
| CST, µm, mean±SD | 378±160 | 365±162 | 0.34 |
| Combined with PDT (%) | 11.7 | 8.7 | 0.16 |
| After COVID-19 pandemic | 45% | 40% | 0.17 |
| BCVA at 3M, logMAR, mean±SD, assigned eye | 0.37±0.40 | 0.27±0.37 | <0.001 |
| CST at 3M, µm, mean±SD | 263±121 | 249±99 | 0.049 |
| BCVA at 1Y, logMAR, mean±SD, assigned eye | 0.35±0.42 | 0.26±0.40 | <0.001 |
| CST at 1Y, µm, mean±SD | 280±150 | 254±108 | 0.0045 |
| Number of injections within 1Y, mean±SD | 4.7±2.6 | 5.2±2.6 | 0.013 |
| Number of visits within first year, mean±SD | 12.3±4.6 | 12.9±4.6 | 0.11 |
| Changes in visual acuity between initial visit and 3M, mean±SD | −0.078±0.31 | −0.90±0.23 | 0.33 |
| Changes in CST between initial visit and 3M, µm, mean±SD | −112±138 | −115±139 | 0.63 |
| Changes in visual acuity between initial visit and 1Y, mean±SD | −0.10±0.35 | −0.10±0.29 | 0.86 |
| Changes in CST between initial visit and 1Y, µm, mean±SD | −98±192 | −108±151 | 0.69 |
| Changes in visual acuity between 3M and 1Y, mean±SD | −0.021±0.22 | −0.011±0.20 | 0.21 |
| Changes in CST between 3M and 1Y, µm, mean±SD | 15±132 | 6.0±94 | 0.50 |

nAMD, neovascular age-related macular degeneration; BCVA, best-corrected visual acuity; CST, central subfoveal thickness; ILM–RPE, inner limiting membrane–retinal pigment epithelium; logMAR, logarithm of the minimum angle of resolution; PCV, polypoidal choroidal vasculopathy; PDT, photodynamic therapy; RAP, retinal angiomatous proliferation; SD, standard deviation; VEGF, vascular endothelial growth factor.

**Table 4. Comparison of the reasons for loss to follow-up at 3 months, 1 year, and 2 years after treatment initiation.**

| | Factors | Odds ratio | 95% CI | p-value |
|---|---|---|---|---|
| 3 months | Greater CST | 1.002 | 1.001-1.003 | <0.0001 |
| | Presence of prior treatment history | 2.41 | 1.56-3.71 | 0.0002 |
| 3 months to 1 year | Worse baseline BCVA | 2.19 | 1.63-2.95 | <0.0001 |
| | Treatment combined with PDT at baseline | 1.82 | 1.19-2.79 | 0.0081 |
| | After COVID-19 pandemic | 1.59 | 1.2-2.11 | 0.0016 |
| 1 year to 2 years | Worse BCVA at 3 months | 1.97 | 1.38-2.79 | 0.0002 |

BCVA, best-corrected visual acuity; CI, confidence interval; CST, central subfoveal thickness; PDT, photodynamic therapy; COVID-19, coronavirus disease 2019.

of 2.19 (95% CI 1.63–2.95) for worse BCVA at baseline, 1.82 (95% CI 1.19–2.79) for combined treatment with PDT, and 1.59 (95% CI 1.20–2.11) for after COVID-19. A significant reason for LTFU between 1 year and 2 years was the BCVA at 3 months after treatment initiation. Logistic regression analysis for each factor revealed an OR of 1.97 (95% CI 1.38–2.79) for worse BCVA at 3 months.

### Comparison of the effects of age, baseline BCVA and nAMD subtype for LTFU between 3 months, 1 year, and 2 years after treatment initiation

Fig 2 shows the distribution of age, baseline BCVA, and nAMD subtype for patients who were LTFU according to the respective follow-up period. Older patients had a higher rate of LTFU. However, patients aged younger than 60 years also had a relatively high LTFU rate by 3 months. When the follow-up period became longer, older patients had higher LTFU rates. Patients with a baseline BCVA logMAR of >;1 had the highest LTFU rate. Subtype of nAMD at baseline did not affect the follow-up rate; however, patients with retinal angiomatous proliferation had a relatively lower LTFU rate during the 2 years after initiation of treatment.

Table 5 shows the results of multivariable logistic regression analysis of characteristics associated with LTFU. The reference category for age and BCVA at baseline was set as the most frequent category from Fig 2. The OR for being LTFU within 3 months was highest for those aged 90–99 years (OR 3.12, 95% CI 1.68–5.79, p = 0.0003), followed by the group for those aged 50–59 years (OR 2.26, 95% CI 1.25–4.08, p = 0.0068), compared to the reference for those aged 70–79 years. Patients with a baseline BCVA (logMAR) of >1 had a higher OR for being LTFU (OR 2.06, 95% CI 1.29–3.29, p = 0.0025) than those with a baseline BCVA of 0.1–0.4, and conversely a lower risk at a baseline BCVA of <0.1 (OR 0.62, 95% CI 0.39–0.98, p = 0.04).

The OR for being LTFU within 1 year was higher at 80–89 years (OR 1.49, 95% CI 1.09–2.04, p = 0.0013) and 90–99 years (OR 2.48, 95% CI 1.35–4.56, p = 0.0035) than at 70–79 years. Patients with a baseline BCVA (logMAR) of >1 had a higher (OR 2.69, 95% CI 1.78–4.07, p ≤ 0.0001) for LTFU than those with a baseline BCVA of 0.1–0.4.

The OR for becoming LTFU within 2 years was higher at 80–89 years (OR 1.96, 95% CI 1.38–2.77, p = 0.0001) and 90–99 years (OR 3.56, 95% CI 1.68–7.54, p = 0.00099) when compared with 70–79 years. Patients with a baseline BCVA (logMAR) of <0.1 had a lower OR (OR 0.65, 95% CI 0.45–0.95, p = 0.028) than those with a baseline BCVA of 0.1–0.4.

### Discussion

This multicenter retrospective study examined the characteristics of patients with nAMD who were LTFU in the early to medium term after initiation of treatment. The study included 2389 patients from 16 specialist retinal facilities across Japan and showed that the LTFU rates were 6.8% within 3 months after initiation of treatment, 13.8% between 3 months and 1 year, and 21.2% between 1 and 2 years.

The common factors for discontinuation of treatment in any period were older age and worse BCVA at baseline, as previously reported [7]. Elderly patients may have more comorbidities and travel-related restrictions on hospital visits [7,8], and worse BCVA at baseline could lead to discontinuation of treatment because of unsatisfactory outcomes and the burden of follow-up. Older patients may also have worse BCVA.

Additional contributors to early LTFU by 3 months were a prior treatment history, unilateral nAMD, and a greater CST. Stepwise regression analysis showed that a prior treatment history and a greater CST were important contributors to LTFU by 3 months. Having a prior treatment history implies long-standing disease with a relatively worse BCVA. A patient who had been started on anti-VEGF treatment by a local clinic and referred to a J-CREST institution might opt for continuation of follow-up at the local clinic if they felt unsatisfied with the results of initial treatment or had difficulties attending for regular visits. A greater CST also suggests progression of disease, which is likely to be less responsive to treatment, such that patients would not have been satisfied with the early response. Interestingly, detailed evaluation showed that patients aged younger than 60 years and those with relatively worse BCVA at baseline tended to discontinue treatment within 3 months. Younger patients could encounter difficulties with regular follow-up because of work commitments, relatively greater costs, possibly better baseline BCVA, and a more favorable initial response.

In terms of risk factors for LTFU in the medium term, namely, between 3 months and 1 year, in addition to the reasons for early LTFU, treatment combined with PDT overlap with the COVID-19 pandemic, and worse BCVA at 3 months were

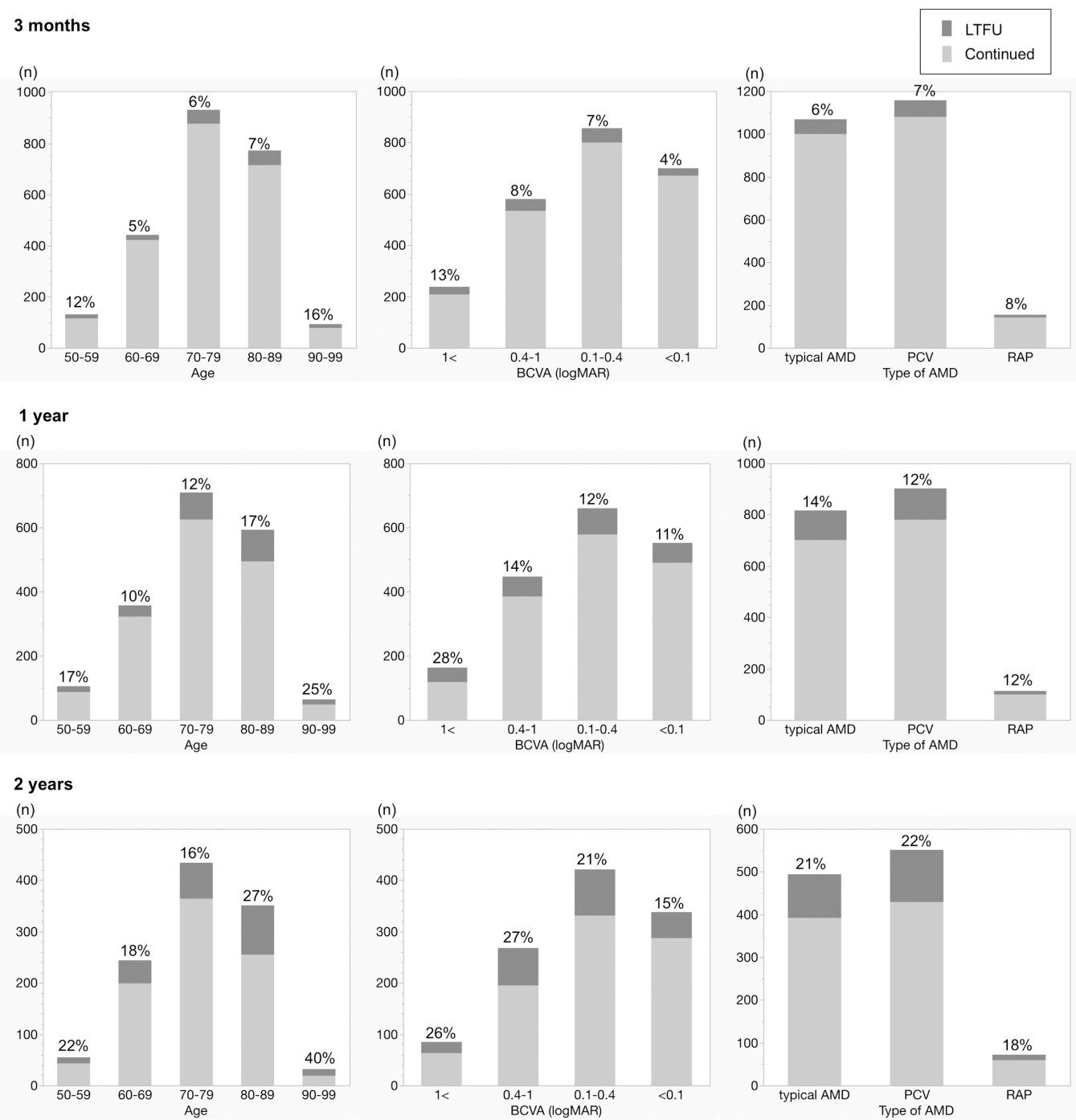

**Fig 2. The distribution of age, baseline BCVA and nAMD subtype for loss to follow-up between 3 months, 1 year, and 2 years after treatment initiation.** BCVA, best-corrected visual acuity; nAMD, neovascular age-related macular degeneration; LTFU, loss to follow-up; PCV, polypoidal choroidal vasculopathy; RAP, retinal angiomatous proliferation.

Table 5. Multivariable logistic regression analysis of characteristics of age and baseline BCVA associated with loss to follow-up.

| 3 months | | | | 1 year | | | | 2 years | | | |
|---|---|---|---|---|---|---|---|---|---|---|---|
| Factors | Odds ratio | 95% CI | p | Factors | Odds ratio | 95% CI | p | Factors | Odds ratio | 95% CI | p |
| Age | | | | Age | | | | Age | | | |
| 50-59 | 2.26 | 1.25-4.08 | *0.0068* | 50-59 | 1.54 | 0.88-2.68 | 0.13 | 50-59 | 1.45 | 0.73-2.89 | 0.29 |
| 60-69 | 0.77 | 0.46-1.30 | 0.33 | 60-69 | 0.81 | 0.53-1.23 | 0.32 | 60-69 | 1.18 | 0.78-1.78 | 0.44 |
| 70-79 | 1 | (Reference) | | 70-79 | 1 | (Reference) | | 70-79 | 1 | (Reference) | |
| 80-89 | 1.29 | 0.88-1.90 | 0.19 | 80-89 | 1.49 | 1.09-2.04 | *0.0013* | 80-89 | 1.96 | 1.38-2.77 | *0.0001* |
| 90-99 | 3.12 | 1.68-5.79 | *0.0003* | 90-99 | 2.48 | 1.35-4.56 | *0.0035* | 90-99 | 3.56 | 1.68-7.54 | *0.0009* |
| BCVA (logMAR) | | | | BCVA (logMAR) | | | | BCVA (logMAR) | | | |
| 1< | 2.06 | 1.29-3.29 | *0.0025* | 1< | 2.69 | 1.78-4.07 | *<0.0001* | 1< | 1.28 | 0.75-2.20 | 0.36 |
| 0.4-1 | 1.23 | 0.82-1.85 | 0.32 | 0.4-1 | 1.14 | 0.80-1.62 | 0.48 | 0.4-1 | 1.38 | 0.96-1.97 | 0.078 |
| 0.1-0.4 | 1 | (Reference) | | 0.1-0.4 | 1 | (Reference) | | 0.1-0.4 | 1 | (Reference) | |
| <0.1 | 0.62 | 0.39-0.98 | *0.04* | <0.1 | 0.89 | 0.63-1.27 | 0.52 | <0.1 | 0.65 | 0.45-0.95 | *0.028* |

CI, confidence interval; BCVA, best-corrected visual acuity; logMAR, logarithm of the minimum angle of resolution

identified. Stepwise regression analysis identified a worse baseline BCVA, treatment combined with PDT, and overlap with the COVID-19 pandemic period to be significant factors associated with LTFU during this period. PDT combined with anti-VEGF treatment could be chosen for eyes with PCV and/or pachychoroid neovasculopathy to achieve earlier stabilization [9–11]. Earlier resolution of fluid after PDT can also facilitate discontinuation of further treatment by both patients and clinicians.

Factors associated with LTFU during the second year were male sex, worse BCVA in the fellow eye at baseline, worse BCVA and greater CST at 3 months and 1 year, and fewer anti-VEGF injections within the first year, in addition to older age and worse BCVA at baseline. The lack of an association of a prior treatment history with LTFU in the second year suggests a disease status requiring continuous treatment regardless of the duration of disease. It is interesting that women seem to continue treatment for longer than men, and one of the possible reasons is a difference in disease subtype by sex; in women, RAP, which often develops bilaterally and requires regular treatment, is more prevalent, while PCV, which requires less regular treatment, is less prevalent [12,13]. It is unfortunate that worse BCVA in the fellow eye can cause discontinuation of treatment after 1 year, possibly because of the need for family support for transport to hospital [14]. Having fewer anti-VEGF injections within 1 year suggests that the patients may have refused regular ongoing injections, resulting in an inadequate response, which could have led to LTFU. A previous study found that patients with longer intervals between injections were more likely to discontinue treatment [15].

Logistic regression analysis showed a significant association of BCVA at 3 months after treatment initiation with LTFU during the second year. Worse BCVA at 3 months could reflect worse baseline BCVA as well as less gain of BCVA after initial treatment. Conversely, patients with better BCVA at baseline could continue treatment for more than 2 years. Earlier diagnosis and initiation of treatment can encourage patients not discontinuing treatment to perceive the benefits of maintaining their BCVA.

It should be noted that the COVID-19 pandemic had a considerable effect on LTFU in patients within 1 year after treatment initiation. There have been many reports of the negative effect of the COVID-19 pandemic on patients with nAMD, including a worsening visual prognosis as a result of long intervals between visits, lack of appropriate examinations, and failure to attend for anti-VEGF injections [16–19]. Our present findings suggest that even during the pandemic, in the very early treatment period, patients would come to hospital with a positive expectation of a response, and that after 1 year, examination and treatment could become routine when the necessity of treatment is understood. Patients in the relatively early treatment phase could easily discontinue treatment in response to external factors.

Treatment of nAMD requires regular clinic visits and anti-VEGF therapy, which impose significant financial and time burdens for patients and caregivers [20,21]. Previous research in Japanese patients with nAMD identified the top five reasons for treatment discontinuation to be no perceived change in vision, financial burden, perceived lack of treatment efficacy, uncertainty about continuing treatment, and long hospital waiting times [22]. These findings align with our present findings that patient dissatisfaction, advanced disease, and lack of a favorable response can cause LTFU. However, the possibility of financial and physical burdens was not investigated.

This study has several limitations. First, it had a multicenter retrospective design, which raises the possibility of bias in treatment and follow-up strategies selected by the treating physicians. Second, LTFU in this study did not necessarily mean cessation of treatment because some patients could have continued their treatment at their original clinic. Third, the study was conducted solely in Japan, where healthcare access and cultural factors may differ from those in other regions. This limits the generalizability of the findings of this study to populations in other countries.

## Conclusion

This study has highlighted the time-specific factors associated with LTFU in patients with nAMD. A worse BCVA at baseline and at 3 months had a serious impact on LTFU in the early and medium term. It is obvious that earlier diagnosis and initiation of treatment before disease progression are essential for obtaining long-term treatment benefits and maintaining patient satisfaction. Not only baseline disease status but also the other factors associated with LTFU elucidated in the present study should be borne in mind to reduce the burdens on patients and prevent further vision loss.

## Acknowledgments

The authors who met the criteria for authorship as listed by the ICMJE in the J-CREST group were Dr. Takaaki Sugisawa, Dr. Fumi Gomi, Dr. Yuri Harada, Dr. Hiroko Imaizumi, Dr. Shuichiro Aoki, Dr. Akiko Miki, Dr. Maya Kishi, Dr. Tomofusa Yamauchi, Dr. Daisuke Nagasato, Dr. Yoko Ozawa, Dr. Masatoshi Dr. Haruta, Dr. Nobuhiro Kato, Dr. Hisashi Matsubara, Dr. Tsutomu Yasukawa, Dr. Aki Kato, Dr. Hiroto Terasaki, Dr. Takao Hirano, Dr. Yasuhiro Iesato, Dr. Hiroki Tsujinaka, Dr. Tomoya Murakami, Dr. Yoshinori Mitamura, Dr. Makiko Wakuta, Dr. Kazuhiro Kimura, Dr. Masahiko Shimura.

We thank Susan Albrecht from Edanz (https://jp.edanz.com/ac) for editing a draft of this manuscript.

## Author contributions

**Conceptualization:** Fumi Gomi.

**Data curation:** Yuri Harada, Hiroko Imaizumi, Shuichiro Aoki, Akiko Miki, Maya Kishi, Tomofusa Yamauchi, Daisuke Nagasato, Yoko Ozawa, Masatoshi Haruta, Nobuhiro Kato, Hisashi Matsubara, Tsutomu Yasukawa, Aki Kato, Hiroto Terasaki, Takao Hirano, Yasuhiro Iesato, Hiroki Tsujinaka, Tomoya Murakami, Yoshinori Mitamura, Makiko Wakuta, Kazuhiro Kimura, Masahiko Shimura.

**Supervision:** Fumi Gomi.

**Writing – original draft:** Takaaki Sugisawa.

**Writing – review & editing:** Fumi Gomi.

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
