## [Decision Letter · Decision Letter 0]

7 Jan 2025

Dear Dr. Gomi,

Thank you for submitting your manuscript to PLOS ONE. After careful consideration, we feel that it has merit but does not fully meet PLOS ONE’s publication criteria as it currently stands. Therefore, we invite you to submit a revised version of the manuscript that addresses the points raised during the review process.

We look forward to receiving your revised manuscript.

Kind regards,

Daniel Duck-Jin Hwang

Academic Editor

PLOS ONE

Journal Requirements:

2. One of the noted authors is a group or consortium ” J-CREST Japan Clinical REtina STudy Group” In addition to naming the author group, please list the individual authors and affiliations within this group in the acknowledgments section of your manuscript. Please also indicate clearly a lead author for this group along with a contact email address.

**Additional Editor Comments:**

This study investigates factors contributing to the loss to follow-up in Japanese patients with neovascular age-related macular degeneration (nAMD) receiving anti-vascular endothelial growth factor (VEGF) therapy. Using a multicenter retrospective case-control design, the researchers analyzed data from 2,389 patients across 16 retinal care facilities in Japan between 2017 and 2020. The study emphasizes the need for targeted interventions to improve adherence, particularly among high-risk groups.

The paper addresses a critical issue in managing nAMD, as loss to follow-up significantly affects treatment outcomes. The use of a large, multicenter dataset adds robustness to the findings, and the identification of time-specific risk factors provides actionable insights for clinicians. However, there are areas for improvement and additional exploration to strengthen the study's contributions.

1. The abstract needs to be rewritten. Currently, it mentions only the results of the multiple regression analysis, where different factors are associated at 3 months (greater CRT, prior treatment history), 1 year (baseline VA), and 2 years (VA at 1 year, age). However, these results are not emphasized in the main text, and the first part of the discussion only mentions older age and baseline VA. It is essential to decide whether the focus will be on highlighting the factors differentiating the ‘Lost to FU’ group and the ‘Continued FU’ group or emphasizing the factors associated with FU loss identified through multiple analysis. The main text should then be revised for a consistent narrative.

2. The results of the multiple regression analysis should be summarized in a single table. Currently, the results are scattered throughout the text, making it difficult for readers to follow. Additionally, instead of vaguely mentioning factors like "CST" or "VA," specify the exact nature of these variables. For example, in "Logistic regression analysis for each factor revealed an OR of 2.19 (95% CI 1.63–2.96) for visual acuity at the initial visit and an OR of 1.82 (95% CI 1.19–2.79) for no use of PDT," it is unclear whether good VA or poor VA is associated with FU loss. Similarly, while PDT is not mentioned in the abstract, the text does not clarify whether PDT use or non-use is associated with FU loss. Such ambiguities are present throughout the manuscript. Ensure precise and clear descriptions of the results.

3. Figure 3, 5, 7

As you know, ‘distribution’ and ‘ratio’ are distinct concepts. For instance, if there are 40 patients aged <60 and 100 patients aged ≥80 at baseline, and 30 of the <60 group and 50 of the ≥80 group experience FU loss, the number of FU loss patients is higher in the ≥80 group (50 vs. 30). However, the FU loss ‘ratio’ is higher in the <60 group (75% vs. 50%). Regarding your analysis, in Figure 3, you state, “Patients with a baseline decimal BCVA of 0.3–0.5 (logMAR 0.30–0.52) had the highest loss to follow-up.” Is this accurate? While VA 0.3–0.5 may have the highest patient distribution, for this group to have the “highest loss,” the ratio of FU loss within this group must also be the highest. However, based on the upper and lower bar graphs in Figure 3, the group with VA <0.1 has the highest FU loss ratio. Such errors are evident in the results for age and VA across Figures 3, 5, and 7. Please calculate the ratio of FU loss for each subgroup in these figures, present them alongside the current data, and revise the text accordingly.

4. Tables 1, 2, 3

Check and revise any discrepancies between the main text and the tables.

For example:

Table 1: The fellow eyes show no difference in VA between groups, while the study eyes show a difference. However, this is incorrectly described in the text. Additionally, the 'Lost to FU' group demonstrates lower bilateral involvement, which should also be mentioned.

Table 2, 3: Specify whether the VA and CST results are for the fellow eyes or the assigned eyes.

5. Statistical analysis

In the discussion, you state, “Multivariate analysis identified a prior treatment history, initial treatment that included PDT, overlap with the COVID-19 pandemic period, and BCVA at 3 months to be factors associated with loss to follow-up in the first year, in addition to age and baseline BCVA.” This appears to reference Table 2, but it is an incorrect description. Table 2 only shows factors with P-value differences between the two groups; it does not represent multivariate analysis. To conduct multiple regression analysis, further analysis must be performed based on these results, as multiple regression is distinct from multivariate analysis. The latter generally requires multiple dependent variables.

Additionally, the Statistical Analysis section in the Methods lacks any mention of the multiple regression analysis or other statistical methods claimed to have been performed. Provide a clear and detailed description of the statistical methods used in your analysis.

6. Limited Generalizability: The study is conducted solely in Japan, where healthcare access and cultural factors may differ from other regions. This limits the generalizability of the findings to global populations.

7. Incomplete Characterization of nAMD Subtypes: While the study mentions different nAMD subtypes, it does not analyze follow-up loss in detail for each subtype. Subtype-specific characteristics might influence adherence rates and treatment responses differently.

I believe this is an excellent study that reflects the significant time and effort contributed by many researchers across various institutions. I hope the two reviewers and my comments will be helpful in further enhancing the quality of this valuable work.

Reviewers' comments:

Reviewer's Responses to Questions

**Comments to the Author**

1. Is the manuscript technically sound, and do the data support the conclusions?

Reviewer #1: Partly

Reviewer #2: Yes

2. Has the statistical analysis been performed appropriately and rigorously?

Reviewer #1: Yes

Reviewer #2: Yes

3. Have the authors made all data underlying the findings in their manuscript fully available?

Reviewer #1: Yes

Reviewer #2: Yes

4. Is the manuscript presented in an intelligible fashion and written in standard English?

Reviewer #1: Yes

Reviewer #2: Yes

Reviewer #1: This study by Sugisawa et al. investigates the factors contributing to loss to follow-up among patients undergoing anti-VEGF therapy for neovascular age-related macular degeneration (nAMD) in 16 Japanese retinal facilities. The manuscript addresses a relevant clinical issue, but several aspects require clarification and improvement to enhance its impact and scientific validity.

1. The term "loss to follow-up" is insufficiently defined. Does it strictly refer to patients who do not return for care, or does it also include those transitioning to other care facilities? Transitioning to local clinics or other centers may not constitute a true loss, as patients could still be receiving treatment. Clear criteria are essential.

2. While the retrospective design is understandable, potential biases in treatment strategies across the 16 centers must be addressed. Treatment approaches (e.g., "as needed" vs. "treat and extend") and medication choices (e.g., bevacizumab, aflibercept, ranibizumab) may vary significantly among centers. Were there notable inter-center differences in management strategies?

3. While the authors focus on ophthalmologic variables such as visual acuity and macular thickness, it is crucial to recognize that factors influencing "loss to follow-up"—such as socioeconomic status, travel distance, drug cost, insurance coverage, and caregiver presence—are likely more significant determinants in this context. Can the authors add those variables directly or indirectly into their analysis?

4. The manuscript acknowledges the COVID-19 pandemic as a factor but lacks a thorough analysis. Subgroup analysis comparing pre-pandemic and pandemic-era cohorts would offer valuable insights into its influence on follow-up rates.

5. The discussion attributes loss to follow-up to factors like patient dissatisfaction or disease severity without direct supporting evidence. Incorporating patient-reported outcomes or survey data could substantiate these claims. Is there existing evidence to support the authors' interpretations?

6. While the authors attempt to draw clinical implications, they fall short of providing actionable recommendations. For instance, how can physicians address loss to follow-up in older patients or those with worse baseline visual acuity or greater central retinal thickness? Practical strategies or interventions would enhance the manuscript's utility.

7. Ensure uniform use of terms, such as "visual acuity" and "BCVA," throughout the manuscript to maintain clarity and readability.

8. Figures 3, 5, and 7 require improvement in axis labels and font sizes for better readability. Standardizing these elements will enhance the overall presentation quality.

Reviewer #2: This paper, which compiles multi-center retrospective results from the Japanese Retinal Society(J-CREST), holds significant meaning. However, there are limitations in its real-wide reflection, as some of the included patient results may not truly represent treatment discontinuation. Nevertheless, considering the importance of long-term treatment for this condition, the publication has value in providing sufficient awareness and caution.

**Do you want your identity to be public for this peer review?** For information about this choice, including consent withdrawal, please see our Privacy Policy

Reviewer #1: No

Reviewer #2: No

---

## [Author Response · Author response to Decision Letter 1]

17 Apr 2025

Comment 1: Please ensure that your manuscript meets PLOS ONE's style requirements, including those for file naming. The PLOS ONE style templates can be found at

Response: We thank the reviewer for this advice and have ensured that the article is correctly formatted.

Comment 2: One of the noted authors is a group or consortium ” J-CREST Japan Clinical REtina STudy Group” In addition to naming the author group, please list the individual authors and affiliations within this group in the acknowledgments section of your manuscript. Please also indicate clearly a lead author for this group along with a contact email address.

Response: We have added the authors’ names in the Acknowledgements section as requested.

Comment 1. The abstract needs to be rewritten. Currently, it mentions only the results of the multiple regression analysis, where different factors are associated at 3 months (greater CRT, prior treatment history), 1 year (baseline VA), and 2 years (VA at 1 year, age). However, these results are not emphasized in the main text, and the first part of the discussion only mentions older age and baseline VA. It is essential to decide whether the focus will be on highlighting the factors differentiating the ‘Lost to FU’ group and the ‘Continued FU’ group or emphasizing the factors associated with FU loss identified through multiple analysis. The main text should then be revised for a consistent narrative.

Response: We appreciate these suggestions and have revised the abstract accordingly, focusing on factors associated with loss to follow-up. We have also addressed these issues in the revised Results and Discussion sections in the main text.

Comment 2: The results of the multiple regression analysis should be summarized in a single table. Currently, the results are scattered throughout the text, making it difficult for readers to follow. Additionally, instead of vaguely mentioning factors like "CST" or "VA," specify the exact nature of these variables. For example, in "Logistic regression analysis for each factor revealed an OR of 2.19 (95% CI 1.63–2.96) for visual acuity at the initial visit and an OR of 1.82 (95% CI 1.19–2.79) for no use of PDT," it is unclear whether good VA or poor VA is associated with FU loss. Similarly, while PDT is not mentioned in the abstract, the text does not clarify whether PDT use or non-use is associated with FU loss. Such ambiguities are present throughout the manuscript. Ensure precise and clear descriptions of the results.

Response: We thank the reviewer for these observations. We have prepared a new Table 4, which summarizes the results of our stepwise regression analysis by time from treatment initiation and deleted the original figures labelled 2, 4, and 6. We apologize for the previously incorrect reporting of the results of our analyses. We have reanalyzed the data and confirm that the final results are correct. We believe this table will help readers to understand the time-specific risk factors for loss to follow-up.

Comment 3: Figure 3, 5, 7 As you know, ‘distribution’ and ‘ratio’ are distinct concepts. For instance, if there are 40 patients aged <60 and 100 patients aged ≥80 at baseline, and 30 of the <60 group and 50 of the ≥80 group experience FU loss, the number of FU loss patients is higher in the ≥80 group (50 vs. 30). However, the FU loss ‘ratio’ is higher in the <60 group (75% vs. 50%). Regarding your analysis, in Figure 3, you state, “Patients with a baseline decimal BCVA of 0.3–0.5 (logMAR 0.30–0.52) had the highest loss to follow-up.” Is this accurate? While VA 0.3–0.5 may have the highest patient distribution, for this group to have the “highest loss,” the ratio of FU loss within this group must also be the highest. However, based on the upper and lower bar graphs in Figure 3, the group with VA <0.1 has the highest FU loss ratio. Such errors are evident in the results for age and VA across Figures 3, 5, and 7. Please calculate the ratio of FU loss for each subgroup in these figures, present them alongside the current data, and revise the text accordingly.

Response: We appreciate these comments and have prepared a new Figure 2 to show the absolute numbers and ratios by age and distribution of BCVA according to follow-up period, as well as disease subtype, as suggested. We have revised the text accordingly.

Comment 4: Tables 1, 2, 3. Check and revise any discrepancies between the main text and the tables.

For example: Table 1: The fellow eyes show no difference in VA between groups, while the study eyes show a difference. However, this is incorrectly described in the text. Additionally, the 'Lost to FU' group demonstrates lower bilateral involvement, which should also be mentioned. Table 2, 3: Specify whether the VA and CST results are for the fellow eyes or the assigned eyes.

Response: We apologize for these reporting errors and have corrected them.

Comment 5: Statistical analysis

In the discussion, you state, “Multivariate analysis identified a prior treatment history, initial treatment that included PDT, overlap with the COVID-19 pandemic period, and BCVA at 3 months to be factors associated with loss to follow-up in the first year, in addition to age and baseline BCVA.” This appears to reference Table 2, but it is an incorrect description. Table 2 only shows factors with P-value differences between the two groups; it does not represent multivariate analysis. To conduct multiple regression analysis, further analysis must be performed based on these results, as multiple regression is distinct from multivariate analysis. The latter generally requires multiple dependent variables.

Additionally, the Statistical Analysis section in the Methods lacks any mention of the multiple regression analysis or other statistical methods claimed to have been performed. Provide a clear and detailed description of the statistical methods used in your analysis.

Response: We have revised our tables and the corresponding text in the revised version of the manuscript. We have also performed using stepwise regression analysis and revised the Methods section accordingly.

Comment 6: Limited Generalizability: The study is conducted solely in Japan, where healthcare access and cultural factors may differ from other regions. This limits the generalizability of the findings to global populations.

Response: We agree with the editor that this is a limitation of the study and have added it in the limitations paragraph in the Discussion section.

Comment 7: Incomplete Characterization of nAMD Subtypes: While the study mentions different nAMD subtypes, it does not analyze follow-up loss in detail for each subtype. Subtype-specific characteristics might influence adherence rates and treatment responses differently.

Response: As shown in Tables 1, 2, and 3, the disease subtype did not affect the follow-up status. However, as the editor points out, patients with type 3 macular neovascularization (pachychoroid neovasculopathy) for example, would be older and more likely to be female, to have bilateral disease and polypoidal choroidal vasculopathy, and would have treatment combined with photodynamic therapy more frequently. We have added the necessary explanatory text in the Discussion section. We have also added a new Figure 2 to make the association of LTFU with nAMD subtype clearer.

I believe this is an excellent study that reflects the significant time and effort contributed by many researchers across various institutions. I hope the two reviewers and my comments will be helpful in further enhancing the quality of this valuable work.

Response: We appreciate these kind words.

REVIEWER 1

This study by Sugisawa et al. investigates the factors contributing to loss to follow-up among patients undergoing anti-VEGF therapy for neovascular age-related macular degeneration (nAMD) in 16 Japanese retinal facilities. The manuscript addresses a relevant clinical issue, but several aspects require clarification and improvement to enhance its impact and scientific validity.

Comment 1: The term "loss to follow-up" is insufficiently defined. Does it strictly refer to patients who do not return for care, or does it also include those transitioning to other care facilities? Transitioning to local clinics or other centers may not constitute a true loss, as patients could still be receiving treatment. Clear criteria are essential.

Response: We agree with the reviewer and have added in the limitations paragraph in the Discussion section some text explaining that loss to follow-up in this study includes cases transitioning to local clinics. We have also addressed the definition of loss to follow-up in the Methods section.

Comment 2: While the retrospective design is understandable, potential biases in treatment strategies across the 16 centers must be addressed. Treatment approaches (e.g., "as needed" vs. "treat and extend") and medication choices (e.g., bevacizumab, aflibercept, ranibizumab) may vary significantly among centers. Were there notable inter-center differences in management strategies?

Response: We thank the reviewer for drawing our attention to this important point. During the study period, the only treatments used were ranibizumab and aflibercept. However, the initial choice of medication varied significantly among institutions, as did follow-up periods and numbers of injections during the respective follow-up periods.

Comment 3: While the authors focus on ophthalmologic variables such as visual acuity and macular thickness, it is crucial to recognize that factors influencing "loss to follow-up"—such as socioeconomic status, travel distance, drug cost, insurance coverage, and caregiver presence—are likely more significant determinants in this context. Can the authors add those variables directly or indirectly into their analysis?

Response: We appreciate these comments and agree with them. In Japan, national health insurance covers 70%–90% of medical fees for all patients but the rate varies according to patient age and income. As the reviewer points out, there are many background factors that contribute to loss to follow-up, including exact drug costs to the patient, travel distance, and availability of a caregiver, as well as the patient’s physical condition. We have added the necessary explanatory text in the Discussion section.

Comment 4: The manuscript acknowledges the COVID-19 pandemic as a factor but lacks a thorough analysis. Subgroup analysis comparing pre-pandemic and pandemic-era cohorts would offer valuable insights into its influence on follow-up rates.

Response: We had compared the patients according to whether their treatment period overlapped with the COVID-19 pandemic or not and revised the Methods section accordingly to make this clear. We have also highlighted the impact of the pandemic on loss to follow-up at a relatively early stage of treatment in the Discussion section.

Comment 5: The discussion attributes loss to follow-up to factors like patient dissatisfaction or disease severity without direct supporting evidence. Incorporating patient-reported outcomes or survey data could substantiate these claims. Is there existing evidence to support the authors' interpretations?

Response: We apologize for this oversight. We have now cited a previous report (reference 14) that discusses the reasons for discontinuation of treatment in patients with nAMD and trust that it is considered at least partially supportive of our present findings.

Comment 6: While the authors attempt to draw clinical implications, they fall short of providing actionable recommendations. For instance, how can physicians address loss to follow-up in older patients or those with worse baseline visual acuity or greater central retinal thickness? Practical strategies or interventions would enhance the manuscript's utility.

Response: We appreciate this valuable comment. As the reviewer would know, it is difficult to find the best strategy for patients with a complex background. However, we agree that physicians should acknowledge these factors and have added some relevant comments in the Conclusions section.

Comment 7: Ensure uniform use of terms, such as "visual acuity" and "BCVA," throughout the manuscript to maintain clarity and readability.

Response: We apologize for any lack of consistency in terms used in the previous version of the manuscript. These have been corrected in the revised version.

Comment 8: Figures 3, 5, and 7 require improvement in axis labels and font sizes for better readability. Standardizing these elements will enhance the overall presentation quality.

Response: We advise that some of the original figures have been deleted in the revised version. However, we have revised the remaining figures as suggested.

REVIEWER 2

This paper, which compiles multi-center retrospective results from the Japanese Retinal Society(J-CREST), holds significant meaning. However, there are limitations in its real-wide reflection, as some of the included patient results may not truly represent treatment discontinuation. Nevertheless, considering the importance of long-term treatment for this condition, the publication has value in providing sufficient awareness and caution.

Response: We appreciate these comments. As the reviewer points out, our study has some limitations, which are included in the limitations paragraph in our revised Discussion section. However, we believe that awareness of time-specific factors affecting follow-up status in patients with nAMD is important. We have revised our original submission according to the suggestions of the editor and reviewers.

---

## [Decision Letter · Decision Letter 1]

22 May 2025

Factors that contribute to loss to follow-up in the medium term after initiation of anti-vascular endothelial growth factor therapy for neovascular age-related macular degeneration in Japanese patients

PONE-D-24-48944R1

Dear Dr. Fumi Gomi,

We’re pleased to inform you that your manuscript has been judged scientifically suitable for publication and will be formally accepted for publication once it meets all outstanding technical requirements.

Kind regards,

Daniel Duck-Jin Hwang

Academic Editor

PLOS ONE

Additional Editor Comments:

The authors have revised and supplemented their manuscript appropriately. Thank you for the diligent revisions made to the manuscript in line with our comments. I truly appreciate your hard work and dedication to improving the manuscript.

Reviewers' comments:

Reviewer's Responses to Questions

**Comments to the Author**

Reviewer #1: All comments have been addressed

2. Is the manuscript technically sound, and do the data support the conclusions?

Reviewer #1: Yes

3. Has the statistical analysis been performed appropriately and rigorously?

Reviewer #1: Yes

4. Have the authors made all data underlying the findings in their manuscript fully available?

Reviewer #1: Yes

5. Is the manuscript presented in an intelligible fashion and written in standard English?

Reviewer #1: Yes

Reviewer #1: All reviewer comments were addressed appropriately, and the quality of the manuscript has improved markedly.

**Do you want your identity to be public for this peer review?** For information about this choice, including consent withdrawal, please see our Privacy Policy

Reviewer #1: No

---

## [Editor Report · Acceptance letter]

PONE-D-24-48944R1

PLOS ONE

Dear Dr. Gomi,

I'm pleased to inform you that your manuscript has been deemed suitable for publication in PLOS ONE. Congratulations! Your manuscript is now being handed over to our production team.

Kind regards,

on behalf of

Professor Daniel Duck-Jin Hwang

Academic Editor

PLOS ONE